

# Interoperability and FAIRness through a novel combination of Web technologies

Mark D. Wilkinson[1], Ruben Verborgh[2], Luiz Olavo Bonino da Silva Santos[3], Tim Clark[4,5], Morris A. Swertz[6], Fleur D.L. Kelpin[6], Alasdair J.G. Gray[7], Erik A. Schultes[8], Erik M. van Mulligen[9], Paolo Ciccarese[10], Arnold Kuzniar[11], Anand Gavai[11], Mark Thompson[12], Rajaram Kaliyaperumal[12], Jerven T. Bolleman[13] and Michel Dumontier[14]

[1] Center for Plant Biotechnology and Genomics UPM-INIA, Universidad Politécnica de Madrid, Madrid, Spain
[2] IMEC, Ghent University, Ghent, Belgium
[3] Dutch Techcentre for Life Sciences, Utrecht, The Netherlands
[4] Department of Neurology, Massachusetts General Hospital, Boston, MA, United States of America
[5] Department of Neurology, Harvard Medical School, Boston, United States of America
[6] Genomics Coordination Center and Department of Genetics, University Medical Center Groningen, Groningen, The Netherlands
[7] Department of Computer Science, School of Mathematical and Computer Sciences, Heriot-Watt University, Edinburgh, United Kingdom
[8] FAIR Data, Dutch TechCenter for Life Science, Utrecht, The Netherlands
[9] Department of Medical Informatics, Erasmus University Medical Center, Rotterdam, The Netherlands
[10] Elmer Innovation Lab, Harvard Medical School, Boston, United States of America
[11] Netherlands eScience Center, Amsterdam, The Netherlands
[12] Department of Human Genetics, Leiden University Medical Center, Leiden, The Netherlands
[13] Swiss-Prot Group, SIB Swiss Institute of Bioinformatics, Centre Medical Universitaire, Geneva, Switzerland
[14] Stanford Center for Biomedical Informatics Research, Stanford University School of Medicine, Stanford, CA, United States of America

Corresponding author
Mark D. Wilkinson,
markw@illuminae.com

## ABSTRACT

Data in the life sciences are extremely diverse and are stored in a broad spectrum of repositories ranging from those designed for particular data types (such as KEGG for pathway data or UniProt for protein data) to those that are general-purpose (such as FigShare, Zenodo, Dataverse or EUDAT). These data have widely different levels of sensitivity and security considerations. For example, clinical observations about genetic mutations in patients are highly sensitive, while observations of species diversity are generally not. The lack of uniformity in data models from one repository to another, and in the richness and availability of metadata descriptions, makes integration and analysis of these data a manual, time-consuming task with no scalability. Here we explore a set of resource-oriented Web design patterns for data discovery, accessibility, transformation, and integration that can be implemented by any general- or special-purpose repository as a means to assist users in finding and reusing their data holdings. We show that by using off-the-shelf technologies, interoperability can be achieved at the level of an individual spreadsheet cell. We note that the behaviours of this architecture compare favourably to the desiderata defined by the FAIR Data Principles, and can therefore represent an exemplar implementation of those principles. The proposed interoperability design patterns may be used to improve discovery and integration of both new and legacy data, maximizing the utility of all scholarly outputs.

## INTRODUCTION

Carefully-generated data are the foundation for scientific conclusions, new hypotheses, discourse, disagreement and resolution of these disagreements, all of which drive scientific discovery. Data must therefore be considered, and treated, as first-order scientific output, upon which there may be many downstream derivative works, among these, the familiar research article (*Starr et al., 2015*). But as the volume and complexity of data continue to grow, a data publication and distribution infrastructure is beginning to emerge that is not *ad hoc*, but rather explicitly designed to support discovery, accessibility, (re)coding to standards, integration, machine-guided interpretation, and re-use.

In this text, we use the word "data" to mean all digital research artefacts, whether they be data (in the traditional sense), research-oriented digital objects such as workflows, or combinations/packages of these (i.e., the concept of a "research object", (*Bechhofer et al., 2013*)). Effectively, all digital entities in the research data ecosystem will be considered data by this manuscript. Further, we intend "data" to include both data and metadata, and recognize that the distinction between the two is often user-dependent. Data, of all types, are often published online, where the practice of open data publication is being encouraged by the scholarly community, and increasingly adopted as a requirement of funding agencies (*Stein et al., 2015*). Such publications utilize either a special-purpose repository (e.g., model-organism or molecular data repositories) or increasingly commonly will utilize general-purpose repositories such as FigShare, Zenodo, Dataverse, EUDAT or even institutional repositories. Special-purpose repositories generally receive dedicated funding to curate and organize data, and have specific query interfaces and APIs to enable exploration of their content. General-purpose repositories, on the other hand, allow publication of data in arbitrary formats, with little or no curation and often very little structured metadata. Both of these scenarios pose a problem with respect to interoperability. While APIs allow mechanized access to the data holdings of a special-purpose repository, each repository has its own API, thus requiring specialized software to be created for each cross-repository query. Moreover, the ontological basis of the curated annotations are not always transparent (neither to humans nor machines), which hampers automated integration. General purpose repositories are less likely to have rich APIs, thus often requiring manual discovery and download; however, more importantly, the frequent lack of harmonization of the file types/formats and coding systems in the repository, and lack of curation, results in much of their content being unusable (*Roche et al., 2015*).

Previous projects, specifically in the bio/medical domain, that have attempted to achieve deep interoperability include caBIO (*Covitz et al., 2003*) and TAPIR (*De Giovanni et al., 2010*). The former created a rich SOAP-based API, enforcing a common interface over all repositories. The latter implemented a domain-specific query language that all participating repositories should respond to. These initiatives successfully enabled

powerful cross-resource data exploration and integration; however, this was done at the expense of broad-scale uptake, partly due to the complexity of implementation, and/or required the unavoidable participation of individual data providers, who are generally resource-strained. Moreover, in both cases, the interoperability was aimed at a specific field of study (cancer, and biodiversity respectively), rather than a more generalized interoperability goal spanning all domains.

With respect to more general-purpose approaches, and where 'lightweight' interoperability was considered acceptable, myGrid (*Stevens, Robinson & Goble, 2003*) facilitated discovery and interoperability between Web Services through rich ontologically-based annotations of the service interfaces, and BioMoby (*Wilkinson et al., 2008*) built on these myGrid annotations by further defining a novel ontology-based service request/response structure to guarantee data-level compatibility and thereby assist in workflow construction (*Withers et al., 2010*). SADI (*Wilkinson, Vandervalk & McCarthy, 2011*), and SSWAP (*Gessler et al., 2009*) used the emergent Semantic Web technologies of RDF and OWL to enrich the machine-readability of Web Service interface definitions and the data being passed—SADI through defining service inputs and outputs as instances of OWL Classes, and SSWAP through passing data embedded in OWL 'graphs' to assist both client and server in interpreting the meaning of the messages. In addition, two Web Service interoperability initiatives emerged from the World Wide Web Consortium—OWL-S (*Martin et al., 2005*) and SAWSDL (*Martin, Paolucci & Wagner, 2007*), both of which used semantic annotations to enhance the ability of machines to understand Web Service interface definitions and operations. All of these Service-oriented projects enjoyed success within the community that adopted their approach; however, the size of these adopting communities have, to date, been quite limited and are in some cases highly domain-specific. Moreover, each of these solutions is focused on Web Service functionality, which represents only a small portion of the global data archive, where most data is published as static records. Service-oriented approaches additionally require data publishers to have considerable coding expertise and access to a server in order to utilize the standard, which further limits their utility with respect to the 'lay' data publishers that make-up the majority of the scholarly community. As such, these and numerous other interoperability initiatives, spanning multiple decades, have yet to convincingly achieve a lightweight, broadly domain-applicable solution that works over a wide variety of static and dynamic source data resources, and can be implemented with minimal technical expertise.

There are many stakeholders who would benefit from progress in this endeavour. Scientists themselves, acting as both producers and consumers of these public and private data; public and private research-oriented agencies; journals and professional data publishers both "general purpose" and "special purpose"; research funders who have paid for the underlying research to be conducted; data centres (e.g., the EBI (*Cook et al., 2016*), and the SIB (*SIB Swiss Institute of Bioinformatics Members, 2016*)) who curate and host these data on behalf of the research community; research infrastructures such as BBMRI-ERIC (*Van Ommen et al., 2015*) and ELIXIR (*Crosswell & Thornton, 2012*), and diverse others. All of these stakeholders have distinct needs with respect to the behaviours of the scholarly data infrastructure. Scientists, for example, need to access research datasets in

order to initiate integrative analyses, while funding agencies and review panels may be more interested in the metadata associated with a data deposition—for example, the number of views or downloads, and the selected license. Due to the diversity of stakeholders; the size, nature/format, and distribution of data assets; the need to support freedom-of-choice of all stakeholders; respect for privacy; acknowledgment of data ownership; and recognition of the limited resources available to both data producers and data hosts, we see this endeavour as one of the *Grand Challenges of eScience*.

In January 2014, representatives of a range of stakeholders came together at the request of the Netherlands eScience Centre and the Dutch Techcentre for Life Sciences (DTL) at the Lorentz Centre in Leiden, the Netherlands, to brainstorm and debate about how to further enhance infrastructures to support a data ecosystem for eScience. From these discussions emerged the notion that the definition and widespread support of a minimal set of community-agreed guiding principles and practices could enable data providers and consumers—machines and humans alike—to more easily find, access, interoperate, and sensibly re-use the vast quantities of information being generated by contemporary data-intensive science. These principles and practices should enable a broad range of integrative and exploratory behaviours, and support a wide range of technology choices and implementations, just as the Internet Protocol (IP) provides a minimal layer that enables the creation of a vast array of data provision, consumption, and visualisation tools on the Internet. The main outcome of the workshop was the definition of the so-called FAIR guiding principles aimed at publishing data in a format that is **Findable**, **Accessible**, **Interoperable** and **Reusable** by both machines and human users. The FAIR Principles underwent a period of public discussion and elaboration, and were recently published (*Wilkinson et al., 2016*). Briefly, the principles state:

**Findable**—data should be identified using globally unique, resolvable, and persistent identifiers, and should include machine-actionable contextual information that can be indexed to support human and machine discovery of that data.

**Accessible**—identified data should be accessible, optimally by both humans and machines, using a clearly-defined protocol and, if necessary, with clearly-defined rules for authorization/authentication.

**Interoperable**—data becomes interoperable when it is machine-actionable, using shared vocabularies and/or ontologies, inside of a syntactically and semantically machine-accessible format.

**Reusable**—Reusable data will first be compliant with the F, A, and I principles, but further, will be sufficiently well-described with, for example, contextual information, so it can be accurately linked or integrated, like-with-like, with other data sources. Moreover, there should be sufficiently rich provenance information so reused data can be properly cited.

While the principles describe the desired features that data publications should exhibit to encourage maximal, automated discovery and reuse, they provide little guidance regarding how to achieve these goals. This poses a problem when key organizations are already endorsing, or even requiring adherence to the FAIR principles. For example, a biological research group has conducted an experiment to examine polyadenylation site

usage in the pathogenic fungus *Magnaporthe oryzae*, recording, by high-throughput 3′-end sequencing, the preference of alternative polyadenylation site selection under a variety of growth conditions, and during infection of the host plant. The resulting data take the form of study-specific Excel spreadsheets, BED alignment graphs, and pie charts of protein functional annotations. Unlike genome or protein sequences and microarray outputs, there is no public curated repository for these types of data, yet the data are useful to other researchers, and should be (at a minimum) easily discovered and interpreted by reviewers or third-party research groups attempting to replicate their results. Moreover, their funding agency, and their preferred scientific journal, both require that they publish their source data in an open public archive according to the FAIR principles. At this time, the commonly used general-purpose data archival resources in this domain do not explicitly provide support for FAIR, nor do they provide tooling or even guidance for how to use their archival facilities in a FAIR-compliant manner. As such, the biological research team, with little or no experience in formal data publishing, must nevertheless self-direct their data archival in a FAIR manner. We believe that this scenario will be extremely common throughout all domains of research, and thus this use-case was the initial focus for this interoperability infrastructure and FAIR data publication prototype.

Here we describe a novel interoperability architecture that combines three pre-existing Web technologies to enhance the discovery, integration, and reuse of data in repositories that lack or have incompatible APIs; data in formats that normally would not be considered interoperable such as Excel spreadsheets and flat-files; or even data that would normally be considered interoperable, but do not use the desired vocabulary standards. We examine the extent to which the features of this architecture comply with the FAIR Principles, and suggest that this might be considered a ''reference implementation'' for the FAIR Principles, in particular as applied to non-interoperable data in any general- or special-purpose repository. We provide two exemplars of usage. The first is focused on a use-case similar to that presented above, where we use our proposed infrastructure to create a FAIR, self-archived scholarly deposit of biological data to the general-purpose Zenodo repository. The second, more complex example has two objectives—first to use the infrastructure to improve transparency and FAIRness of metadata describing the inclusion criterion for a dataset, representing a subset of a special-purpose, curated resource (UniProt); and second, to show how even the already FAIR data within UniProt may be transformed to increase its FAIRness even more by making it interoperable with alternative ontologies and vocabularies, and more explicitly connecting it to citation information. Finally, we place this work in the context of other initiatives and demonstrate that it is complementary to, rather than in competition with, other initiatives.

## METHODS

### Implementation

#### Overview of technical decisions and their justification

The World Wide Web Consortium's (W3C) Resource Description Framework (RDF) offers the ability to describe entities, their attributes, and their relationships with explicit

semantics in a standardized manner compatible with widely used Web application formats such as JSON and XML. The Linked Data Principles (*Berners-Lee, 2006*) mandate that data items and schema elements are identified by HTTP-resolvable URIs, so the HTTP protocol can be used to obtain the data. Within an RDF description, using shared public ontology terms for metadata annotations supports search and large scale integration. Given all of these features, we opted to use RDF as the basis of this interoperability infrastructure, as it was designed to share data on the Web.

Beyond this, there was a general feeling that any implementation that required a novel data discovery/sharing "Platform", "Bus", or API, was beyond the minimal design that we had committed to; it would require the invention of a technology that all participants in the data ecosystem would then be required to implement, and this was considered a non-starter. However, there needed to be some form of coalescence around the mechanism for finding and retrieving data. Our initial target-community—that is, the biomedical sciences—have embraced lightweight HTTP interfaces. We propose to continue this direction with an implementation based on REST (*Fielding & Taylor, 2002*), as several of the FAIR principles map convincingly onto the objectives of the REST architectural style for distributed hypermedia systems, such as having resolvable identifiers for all entities, and a common machine-accessible approach to discovering and retrieving different representations of those entities. The implementation we describe here is largely based on the HTTP GET method, and utilizes rich metadata and hypermedia controls. We use widely-accepted vocabularies not only to describe the data in an interoperable way, but also to describe its nature (e.g., the context of the experiment and how the data was processed) and how to access it. These choices help maximize uptake by our initial target-community, maximize interoperability between resources, and simplify construction of the wide (not pre-defined) range of client behaviours we intend to support.

Confidential and privacy-sensitive data was also an important consideration, and it was recognized early on that it must be possible, within our implementation, to identify and richly describe data and/or datasets without necessarily allowing direct access to them, or by allowing access through existing regulatory frameworks or security infrastructures. For example, many resources within the International Rare Disease Research Consortium participate in the RD Connect platform (*Thompson et al., 2014*) which has defined the "disease card"—a metadata object that gives overall information about the individual disease registries, which is then incorporated into a "disease matrix". The disease matrix provides aggregate data about what disease variants are in the registry, how many individuals represent each disease, and other high-level descriptive data that allows, for example, researchers to determine if they should approach the registry to request full data access.

Finally, it was important that the data host/provider is not *necessarily* a participant in making their data interoperable—rather, the interoperability solution should be capable of adapting existing data with or without the source provider's participation. This ensures that the interoperability objectives can be pursued for projects with limited resourcing, that 'abandoned' datasets may still participate in the interoperability framework, but most importantly, that those with the needs and the resources should adopt the responsibility for making their data-of-interest interoperable, even if it is not owned by them. This distributes

the problem of migrating data to interoperable formats over the maximum number of stakeholders, and ensures that the most crucial resources—those with the most demand for interoperability—become the earliest targets for migration.

With these considerations in mind, we were inspired by three existing technologies whose features were used in a novel combination to create an interoperability infrastructure for both data and metadata, that is intended to also addresses the full range of FAIR requirements. Briefly, the selected technologies are:

(1) The W3C's Linked Data Platform (*Speicher, Arwe & Malhotra, 2015*). We generated a model for hierarchical dataset containers that is inspired by the concept of a Linked Data Platform (LDP) Container, and the LDP's use of the Data Catalogue Vocabulary (DCAT, *Maali, Erickson & Archer, 2014*) for describing datasets, data elements, and distributions of those data elements. We also adopt the DCAT's use of Simple Knowledge Organization System (SKOS, *Miles & Bechhofer, 2009*) Concept Schemes as a way to ontologically describe the content of a dataset or data record.

(2) The RDF MappingLanguage (RML, *Dimou et al., 2014*). RML allows us to describe one or more possible RDF representations for any given dataset, and do so in a manner that is, itself, FAIR: every sub-component of an RML model is Findable, Accessible, Interoperable, and Reusable. Moreover, for many common semi-structured data, there are generic tools that utilize RML models to dynamically drive the transformation of data from these opaque representations into interoperable representations (https://github.com/RMLio/RML-Mapper).

(3) Triple Pattern Fragments (TPF—*Verborgh et al., 2016*). A TPF interface is a REST Web API to retrieve RDF data from data sources in any native format. A TPF server accepts URLs that represent triple patterns [Subject, Predicate, Object], where any of these three elements may be constant or variable, and returns RDF triples from its data source that match those patterns. Such patterns can be used to obtain entire datasets, slices through datasets, or individual data points even down to a single triple (essentially a single cell in a spreadsheet table). Instead of relying on a standardized contract between servers and clients, a TPF interface is self-describing such that automated clients can discover the interface and its data.

We will now describe in detail how we have applied key features of these technologies, in combination, to provide a novel data discoverability architecture. We will later demonstrate that this combination of technologies also enables both metadata and data-level interoperability even between opaque objects such as flat-files, allowing the data within these objects to be queried in parallel with other data on the Semantic Web.

### Metadata interoperability—the "FAIR Accessor" and the linked data platform

The Linked Data Platform "*defines a set of rules for HTTP operations on Web resources…to provide an architecture for read-write Linked Data on the Web*" (https://www.w3.org/TR/ldp/). All entities and concepts are identified by URLs, with machine-readable metadata describing the function or purpose of each URL and the nature of the resource that will be returned when that URL is resolved.

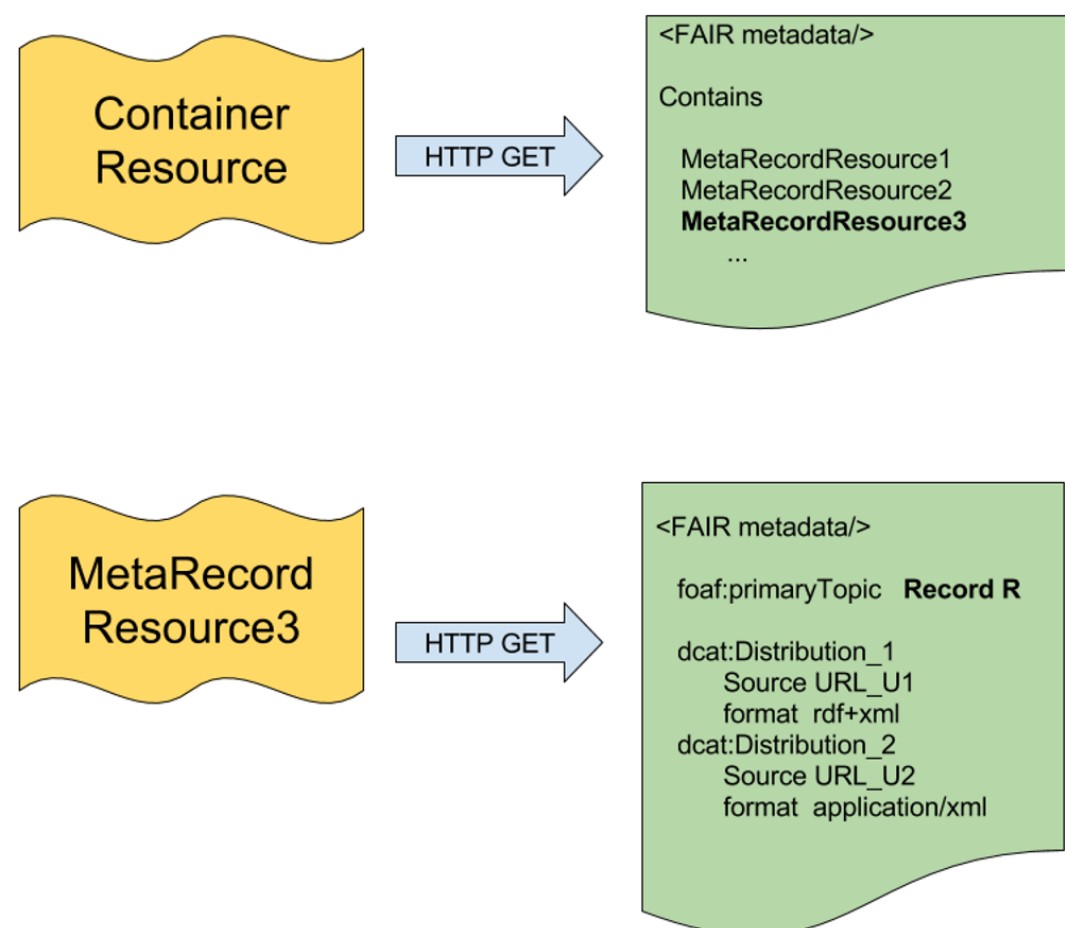

**Figure 1** **The two layers of the FAIR Accessor.** Inspired by the LDP Container, there are two resources in the FAIR Accessor. The first resource is a Container, which responds to an HTTP GET request by providing FAIR metadata about a composite research object, and optionally a list of URLs representing MetaRecords that describe individual components within the collection. The MetaRecord resources resolve by HTTP GET to documents containing metadata about an individual data component and, optionally, a set of links structured as DCAT Distributions that lead to various representations of that data.

Within the LDP specification is the concept of an LDP Container. A basic implementation of LDP containers involves two ''kinds'' of resources, as diagrammed in Fig. 1. The first type of resource represents the container—a metadata document that describes the shared features of a collection of resources, and (optionally) the membership of that collection. This is analogous to, for example, a metadata document describing a data repository, where the repository itself has features (ownership, curation policy, etc.) that are independent from the individual data records within that repository (i.e., the members of the collection). The second type of resource describes a member of the contained collection and (optionally) provides ways to access the record itself.

Our implementation, which we refer to as the ''FAIR Accessor'', utilizes the container concept described by the LDP, however, it does not require a full implementation of LDP, as we only require read functionality. In addition, other requirements of LDP would

have added complexity without notable benefit. Our implementation, therefore, has two resource types based on the LDP Container described above, with the following specific features:

**Container resource:** This is a composite research object (of any kind - repository, repository-record, database, dataset, data-slice, workflow, etc.). Its representation could include scope or knowledge-domain covered, authorship/ownership of the object, latest update, version number, curation policy, and so forth. This metadata may or may not include URLs representing MetaRecord resources (described below) that comprise the individual elements within the composite object. Notably, the Container URL provides a resolvable identifier independent from the identifier of the dataset being described; in fact, the dataset may not have an identifier, as would be the case, for example, where the container represents a dynamically-generated data-slice. In addition, Containers may be published by anyone—that is, the publisher of a Container may be independent from the publisher of the research object it is describing. This enables one of the objectives of our interoperability layer implementation—that anyone can publish metadata about any research object, thus making those objects more FAIR.

**MetaRecord resource**: This is a specific element within a collection (data point, record, study, service, etc.). Its representation should include information regarding licensing and accessibility, access protocols, rich citation information, and other descriptive metadata. It also includes a reference to the container(s) of which it is a member (the Container URL). Finally, the MetaRecord may include further URLs that provide direct access to the data itself, with an explicit reference to the associated data format by its MIME type (e.g., text/html, application/json, application/vnd.ms-excel, text/csv, etc.). This is achieved using constructs from the Data Catalogue Vocabulary (DCAT; *W3C, 2014*), which defines the concept of a data "Distribution", which includes metadata facets such as the data source URL and its format. The lower part of Fig. 1 diagrams how multiple DCAT Distributions may be a part of a single MetaRecord. As with Container resources, MetaRecords may be published by anyone, and independently of the original data publisher.

In summary, the FAIR Accessor shares commonalities with the Linked Data Platform, but additionally recommends the inclusion of rich contextual metadata, based on the FAIR Principles, that facilitate discovery and interoperability of repository and record-level information. The FAIR Accessor is read-only, utilizing only HTTP GET together with widely-used semantic frameworks to guide both human and machine exploration. Importantly, the lack of a novel API means that the information is accessible to generic Web-crawling agents, and may also be processed if that agent "understands" the vocabularies used. Thus, in simplistic terms, the Accessor can be envisioned as a series of Web pages, each containing metadata, and hyperlinks to more detailed metadata and/or data, where the metadata elements and relationships between the pages are explicitly explained to Web crawlers.

To help clarify this component prior to presenting the more complex components of our interoperability proposal, we will now explore our first use case—data self-archival. A simple FAIR Accessor has been published online (*Rodriguez Iglesias et al., 2016*) in the Zenodo general-purpose repository. The data self-archival in this citation

represents a scenario similar to the polyadenylation use-case described in the Introduction section. In this case, the data describes the evolutionary conservation of components of the RNA Metabolism pathway in fungi as a series of heatmap images. The data deposit, includes a file 'RNAME_Accessor.rdf' which acts as the Container Resource. This document includes metadata about the deposit (authorship, topic, etc.), together with a series of 'contains' relationships, referring to MetaRecords inside of the file 'RNAME_Accessor_Metarecords.rdf'. Each MetaRecord is about one of the heatmaps, and in addition to metadata about the image, includes a link to the associated image (datatype image/png) and a link to an RDF representation of the same information represented by that image (datatype application/rdf+xml). It should be noted that much of the content of those Accessor files was created using a text editor, based on template RDF documents. The structure of these two documents are described in more detail in the Results section, which includes a full walk-through of a more complex exemplar Accessor.

At the metadata level, therefore, this portion of the interoperability architecture provides a high degree of FAIRness by allowing machines to discover and interpret useful metadata, and link it with the associated data deposits, even in the case of a repository that provides no FAIR-support. Nevertheless, these components do not significantly enhance the FAIRness and interoperability of the data itself, which was a key goal for this project. We will now describe the application of two recently-published Web technologies—Triple Pattern Fragments and RML—to the problem of data-level interoperability. We will show that these two technologies can be combined to provide an API-free common interface that may be used to serve, in a machine-readable way, FAIR data transformations (either from non-FAIR data, or transformations of FAIR data into novel ontological frameworks). We will also demonstrate how this FAIR data republishing layer can be integrated into the FAIR Accessor to provide a machine-traversable path for incremental drill-down from high-level repository metadata all the way through to individual data points within a record, and back.

### Data interoperability: discovery of compatible data through RML-based FAIR profiles

In our approach to data-level interoperability, we first identified a number of desiderata that the solution should exhibit:

1. View-harmonization over dissimilar datatypes, allowing discovery of *potentially* integrable data within non-integrable formats.
2. Support for a multitude of source data formats (XML, Excel, CSV, JSON, binary, etc.)
3. "Cell-level" discovery and interoperability (referring to a "cell" in a spreadsheet)
4. Modularity, such that a user can make interoperable only the data component of-interest to them
5. Reusability, avoiding "one-solution-per-record" and minimizing effort/waste
6. Must use standard technologies, and reuse existing vocabularies
7. Should not require the participation of the data host (for public data).

The approach we selected was based on the premise that data, in any format, could be metamodelled as a first step towards interoperability; i.e., the salient data-types and

relationships within an opaque data "blob" could be described in a machine-readable manner. The metamodels of two data sources could then be compared to determine if their contained data was, in principle, integrable.

We referred to these metamodels as "FAIR Profiles", and we further noted that we should support multiple metamodels of the same data, differing in structure or ontological/semantic framework, within a FAIR Profile. For example, a data record containing blood pressure information might have a FAIR Profile where this facet is modelled using both the SNOMED vocabulary and the ICD10 vocabulary, since the data facet can be understood using either. We acknowledge that these meta-modelling concepts are not novel, and have been suggested by a variety of other projects such as DCAT and Dublin Core (the DC Application Profile (*Heery & Patel, 2000*)), and have been extensively described by the ISO 11179 standard for "metadata registries". It was then necessary to select a modelling framework for FAIR Profiles capable of representing arbitrary, and possibly redundant, semantic models.

Our investigation into relevant existing technologies and implementations revealed a relatively new, unofficial specification for a generic mapping language called "RDF Mapping Language" (RML *Dimou et al., 2014*). RML is an extension of R2RML (*Das, Sundara & Cyganiak, 2012*), a W3C Recommendation for mapping relational databases to RDF, and is described as "*a uniform mapping formalization for data in different format, which [enables] reuse and exchange between tools and applied data*" (*Dimou et al., 2014*). An RML map describes the triple structure (subject, predicate, object, abbreviated as [S,P,O]), the semantic types of the subject and object, and their constituent URI structures, that would result from a transformation of non-RDF data (of any kind) into RDF data. RML maps are modular documents where each component describes the schema for a single-resource-centric graph (i.e., a graph with all triples that share the same subject). The "object" position in each of these map modules may be mapped to a literal, or may be mapped to another RML module, thus allowing linkages between maps in much the same way that the object of an RDF triple may become the subject of another triple. RML modules therefore may then be assembled into a complete map representing both the structure and the semantics of an RDF representation of a data source. RML maps themselves take the form of RDF documents, and can be published on the Web, discovered, and reused, via standard Web technologies and protocols. RML therefore fulfils each of the desiderata for FAIR Profiles, and as such, we selected this technology as the candidate for their implementation. Comparing with related technologies, this portion of our interoperability prototype serves a similar purpose to the XML Schema (XSD; *Fallside & Walmsley, 2004*) definitions within the output component of a Web Services Description Language (WSDL) document, but unlike XSD, is capable of describing the structure and semantics of RDF graphs.

Of particular interest to us was the modularity of RML—its ability to model individual triples. This speaks directly to our desiderata 4, where we do not wish to require (and should not expect) a modeller to invest the time and effort required to fully model every facet of a potentially very complex dataset. Far more often, individuals will have an interest in only one or a few facets of a dataset. As such, we chose to utilize RML models at their

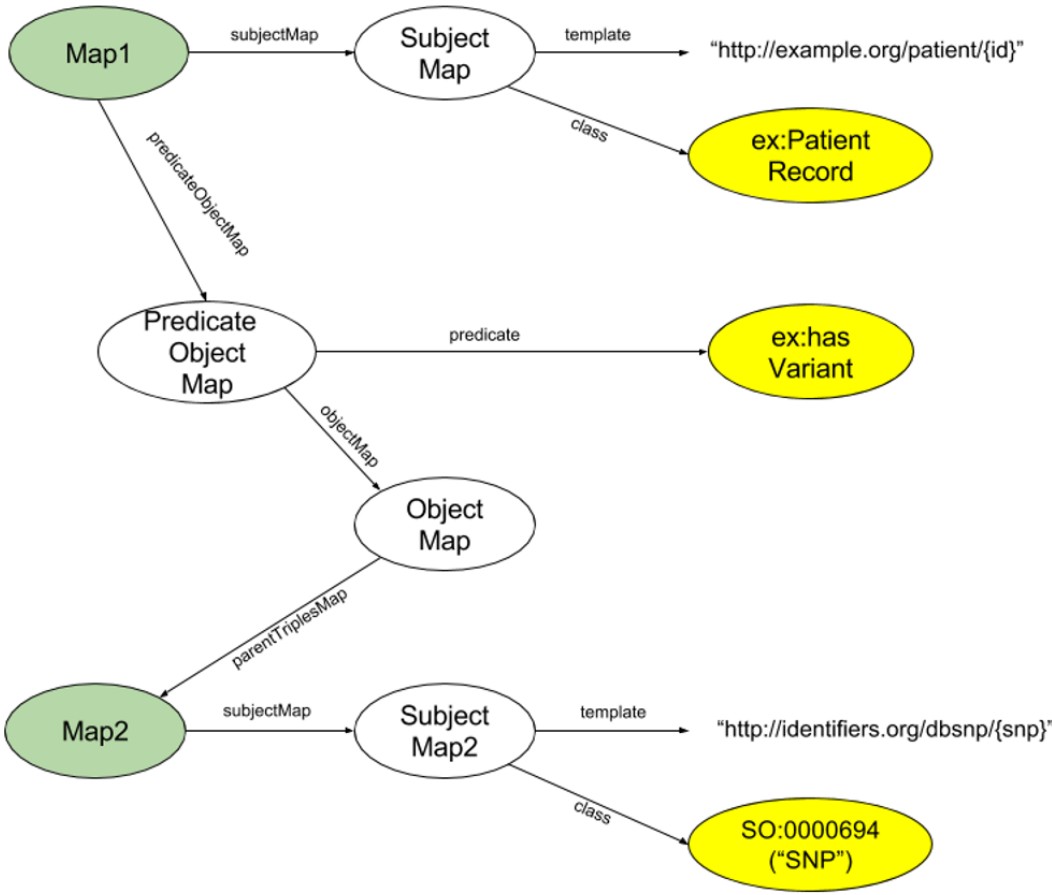

**Figure 2** **Diagram of the structure of an exemplar Triple Descriptor representing a hypothetical record of a SNP in a patient's genome.** In this descriptor, the Subject will have the URL structure *http://example.org/patient/{id}*, and the Subject is of type PatientRecord. The Predicate is hasVariant, and the Object will have URL structure http://identifiers.org/dbsnp/{snp} with the rdf:type from the sequence ontology "0000694" (which is the concept of a "SNP"). The two nodes shaded green are of the same ontological type, showing the iterative nature of RML, and how individual RML Triple Descriptors will be concatenated into full FAIR Profiles. The three nodes shaded yellow are the nodes that define the subject type, predicate and object type of the triple being described.

highest level of granularity—that is, we require a distinct RML model for each triple pattern (subject+type, predicate, object+type) of interest. We call these small RML models "Triple Descriptors". An exemplar Triple Descriptor is diagrammed in Fig. 2. There may be many Triple Descriptors associated with a single data resource. Moreover, multiple Triple Descriptors may model the same facet within that data resource, using different URI structures, subject/object semantic types, or predicates, thus acting as different "views" of that data facet. Finally, then, the aggregation of all Triple Descriptors associated with a specific data resource produces a FAIR Profile of that data. Note that FAIR Profiles are not necessarily comprehensive; however, by aggregating the efforts of all modellers, FAIR Profiles model only the data facets that are most important to the community.

FAIR Profiles enable view harmonization over compatible but structurally non-integrable data, possibly in distinct repositories. The Profiles of one data resource can

be compared to the Profiles of another data resource to identify commonalities between their Triple Descriptors at the semantic level, even if the underlying data is semantically opaque and/or structurally distinct—a key step toward Interoperability. FAIR Profiles, therefore, have utility, independent of any *actuated* transformation of the underlying data, in that they facilitate compatible data discovery. Moreover, with respect to desiderata 5, Triple Descriptors, and sometimes entire FAIR Profiles, are RDF documents published on the Web, and therefore may be reused to describe new data resources, anywhere on the Web, that contain similar data elements, regardless of the native representation of that new resource, further simplifying the goal of data harmonization.

### Data interoperability: data transformation with FAIR projectors and triple pattern fragments

The ability to identify *potentially* integrable data within opaque file formats is, itself, a notable achievement compared to the *status quo*. Nevertheless, beyond just discovery of relevant data, our interoperability layer aims to support and facilitate cross-resource data integration and query answering. This requires that the data is not only semantically described, but is also semantically and syntactically transformed into a common structure.

Having just presented a mechanism to describe the structure and semantics of data—Triple Descriptors in RML—what remains lacking is a way to retrieve data consistent with those Triple Descriptors. We require a means to expose transformed data without worsening the existing critical barrier to interoperability—opaque, non-machine-readable interfaces and API proliferation (*Verborgh & Dumontier, 2016*). What is required is a universally-applicable way of retrieving data generated by a (user-defined) data extraction or transformation process, that does not result in yet another API.

The Triple Pattern Fragments (TPF) specification (*Verborgh et al., 2016*) defines a REST interface for publishing triples. The server receives HTTP GET calls on URLs that contain a triple pattern [S,P,O], where any component of that pattern is either a constant or a variable. In response, a TPF server returns pages with all triples from its data source that match the incoming pattern. As such, any given triple pattern has a distinct URL.

We propose, therefore, to combine three elements—data transformed into RDF, which is described by Triple Descriptors, and served via TPF-compliant URLs. We call this combination of technologies a "FAIR Projector". A FAIR Projector, therefore, is a Web resource (i.e., something identified by a URL) that is associated with both a particular data source, and a particular Triple Descriptor. Calling HTTP GET on the URL of the FAIR Projector produces RDF triples from the data source that match the format defined by that Projector's Triple Descriptor. The originating data source behind a Projector may be a database, a data transformation script, an analytical web service, another FAIR Projector, or any other static or dynamic data-source. Note that we do not include a transformation methodology in this proposal; however, we address this issue and provide suggestions in the Discussion section. There may, of course, be multiple projectors associated with any given data source, serving a variety of triples representing different facets of that data.

### Linking the components: FAIR projectors and the FAIR accessor

At this point, we have a means for requesting triples with a particular structure—TPF Servers—and we have a means of describing the structure and semantics of those triples—Triple Descriptors. Together with a source of RDF data, these define a FAIR Projector. However, we still lack a formal mechanism for linking TPF-compliant URLs with their associated Triple Descriptors, such that the discovery of a Triple Descriptor with the desired semantics for a particular data resource, also provides its associated Projector URL.

We propose that this association can be accomplished, without defining any novel API or standard, if the output of a FAIR Projector is considered a DCAT Distribution of a particular data source, and included within the MetaRecord of a FAIR Accessor. The URL of the Projector, and its Triple Descriptor, become metadata facets of a new dcat:Distribution element in the MetaRecord. This is diagrammed in Fig. 3, where Distribution_3 and Distribution_4 include Triple Pattern Fragment-formatted URLs representing the FAIR Projector, and the Triple Descriptor RML model describing the structure and semantics of the data returned by calling that Projector.

Thus, all components of this interoperability system—from the top level repository metadata, to the individual data cell—are now associated with one another in a manner that allows mechanized data discovery, harmonization, and retrieval, including relevant citation information. No novel technology or API was required, thus allowing this rich combination of data and metadata to be explored using existing Web tools and crawlers.

## RESULTS

In the previous section, we provided the URL to a simple exemplar FAIR Accessor published on Zenodo. To demonstrate the interoperability system in its entirety—including both the Accessor and the Projector components—we will now proceed through a second exemplar involving the special-purpose repository for protein sequence information, UniProt. In this example, we examine a FAIR Accessor to a dataset, created through a database query, that consists of a specific "slice" of the Protein records within the UniProt database—that is, the set of proteins in *Aspergillus nidulans FGSC A4* (NCBI Taxonomy ID 227321) that are annotated as being involved in mRNA Processing (Gene Ontology Accession GO:0006397). We first demonstrate the functionality of the two layers of the FAIR Accessor in detail. We then demonstrate a FAIR Projector, and show how its metadata integrates into the FAIR Accessor. In this example, the Projector modifies the ontological framework of the UniProt data such that the ontological terms used by UniProt are replaced by the terms specified in EDAM—an ontology of bioinformatics operations, datatypes, and formats (*Ison et al., 2013*). We will demonstrate that this transformation is specified, in a machine-readable way, by the FAIR Triple Descriptor that accompanies each Projector's metadata.

### The two-step FAIR accessor

The example FAIR Accessor accesses a database of RDF hosted by UniProt, and issues the following query over that database (expressed in the standard RDF query language SPARQL):

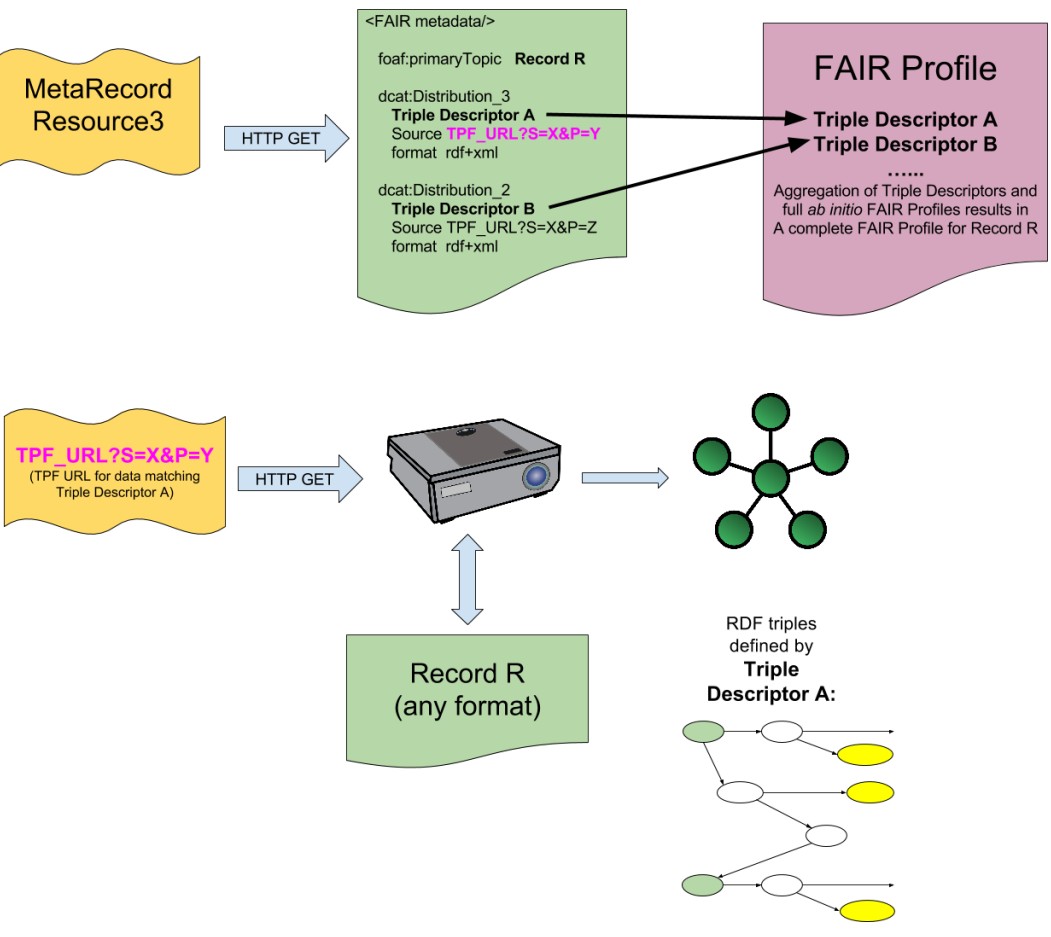

**Figure 3** **Integration of FAIR Projectors into the FAIR Accessor.** Resolving the MetaRecord resource returns a metadata document containing multiple DCAT Distributions for a given record, as in Fig. 1. When a FAIR Projector is available, additional DCAT Distributions are included in this metadata document. These Distributions contain a URL (purple text) representing a Projector, and a Triple Descriptor that describes, in RML, the structure and semantics of the Triple(s) that will be obtained from that Projector resource if it is resolved. These Triple Descriptors may be aggregated into FAIR Profiles, based on the Record that they are associated with (Record R, in the figure) to give a full mapping of all available representations of the data present in Record R.

```
PREFIX up:<http://purl.uniprot.org/core/>
PREFIX taxon:<http://purl.uniprot.org/taxonomy/>
PREFIX rdf:<http://www.w3.org/1999/02/22-rdf-syntax-ns#>
PREFIX GO:<http://purl.obolibrary.org/obo/GO_>
SELECT DISTINCT ?id

WHERE
{
  ?protein a up:Protein ;
  up:organism taxon:227321 ;
  up:classifiedWith/rdfs:subClassOf GO:0006397 .
```

```
    BIND(substr(str(?protein), 33) as ?id)
}
```

Accessor output is retrieved from the Container Resource URL:

`http://linkeddata.systems/Accessors/UniProtAccessor`

The result of calling GET on the Container Resource URL is visualized in Fig. 4, where Tabulator (*Berners-Lee et al., 2006*) is used to render the output as HTML for human-readability.

Of particular note are the following metadata elements:

| | |
|---|---|
| http://purl.org/dc/elements/1.1/license | https://creativecommons.org/licenses/by-nd/4.0/ |
| http://purl.org/pav/authoredBy | http://orcid.org/0000-0002-9699-485X |
| http://rdfs.org/ns/void#entities | 82 |
| a | http://purl.org/dc/dcmitype/Dataset |
| | http://www.w3.org/ns/ldp#BasicContainer |
| | http://www.w3.org/ns/prov#Collection |
| http://www.w3.org/ns/dcat#contactPoint | http://biordf.org/DataFairPort/MiscRDF/Wilkinson.rdf |
| http://www.w3.org/ns/dcat#keyword | "Aspergillus nidulans", "Aspergillus", "Proteins", "RNA Processing"; |
| http://www.w3.org/ns/dcat#theme | http://linkeddata.systems/ConceptSchemes/RNA_Processing_conceptscheme.rdf |
| http://www.w3.org/ns/ldp#contains | http://linkeddata.systems/cgi-bin/Accessors/UniProtAccessor/C8VIL6 |
| | http://linkeddata.systems/cgi-bin/Accessors/UniProtAccessor/C8V2B3 |
| | … |

- License information is provided as an HTML + RDFa document, following one of the primary standard license forms published by Creative Commons. This allows the license to be unambiguously interpreted by both machines and people prior to accessing any data elements, an important feature that will be discussed later.
- Authorship is provided by name, using the Academic Research Project Funding Ontology (ARPFO), but is also unambiguously provided by a link to the author's ORCID, using the Provenance Authoring and Versioning (PAV; *Ciccarese et al., 2013*) ontology.
- The repository descriptor is typed as being a Dublin Core Dataset, a Linked Data Platform container, and a Provenance Collection, allowing it to be interpreted by a variety of client agents, and conforming to several best-practices, such as the Healthcare and Life Science Dataset Description guidelines (*Gray et al., 2015*; *Dumontier et al., 2016*).

| UniProt Slice FAIR Accessor - Aspergillus RNA Processing proteins | creator | wilkinsonlab.info/ |
| | language | eng |
| | license | 4.0/ |
| | title | UniProt Slice FAIR Accessor - Aspergillus RNA Processing proteins |
| | authored By | 0000 0002 9699 485X |
| | version | UniProt release 2016_09 |
| | entities | 82 |
| | term has Principal Investigator | Dr. Mark Wilkinson |
| | type | Dataset |
| | | Basic Container |
| | | Collection |
| | contact Point | Wilkinson.rdf |
| | description | Takes a SPARQL query of the UniProt database specific to proteins and their GO annotations related to RNA Procssing proteins in Aspergillus and makes it a FAIR Accessor source. The query being executed is: |

```
PREFIX up:<http://purl.uniprot.org/core/>
PREFIX taxon:<http://purl.uniprot.org/taxonomy/>
PREFIX rdf:<http://www.w3.org/1999/02/22-rdf-syntax-ns#>
PREFIX GO:<http://purl.obolibrary.org/obo/GO_>
SELECT DISTINCT ?id

WHERE
{
    ?protein a up:Protein ;
    up:organism taxon:227321 ;
    up:classifiedWith/rdfs:subClassOf GO:0006397 .
    BIND(substr(str(?protein), 33) as ?id)
}
```

| | identifier | Uni Prot Accessor |
| | keyword | Aspergillus nidulans |
| | | Aspergillus |
| | | Proteins |
| | | RNA Processing |
| | landing Page | uniprot.org/ |
| | language | en |
| | publisher | wilkinsonlab.info/ |
| | theme | RNA Processing conceptscheme.rdf |
| | contains | C8V1L6 |
| | | C8V2B3 |
| | | C8V609 |
| | | C8V6T1 |
| | | C8V8F9 |
| | | C8V9C2 |
| | | C8V9G4 |
| | | C8VBS9 |
| | | C8VBV2 |

**Figure 4** **A representative portion of the output from resolving the Container Resource of the FAIR Accessor, rendered into HTML by the Tabulator Firefox plugin.** The three columns show the label of the Subject node of all RDF Triples (left), the label of the URI in the predicate position of each Triple (middle), and the value of the Object position (right), where blue text indicates that the value is a Resource, and black text indicates that the value is a literal.

- Contact information is provided in a machine-readable manner via the Friend of a Friend (FoaF) record of the author, and the DCAT ontology "contactPoint" property.
- Human readable keywords, using DCAT, are mirrored and/or enhanced by a machine-readable RDF document which is the value of the DCAT "theme" property. This RDF document follows the structure determined by the Simple Knowledge Organization System (SKOS) ontology, and lists the ontological terms that describe the repository for machine-processing.
- Finally, individual records within the dataset are represented as the value of the Linked Data Platform "contains" property, and provided as a possibly paginated list of URLs (a discussion of machine-actionable pagination will not be included here). These URLs are the MetaRecord Resource URLs shown in Fig. 1.

Following the flow in Fig. 1, the next step in the FAIR Accessor is to resolve a MetaRecord Resource URL. For clarity, we will first show the metadata document that is returned if

| UniProt Protein C8V1L6 | bibliographic Citation | The UniProt Consortium (2015). UniProt: a hub for protein information. Nucleic Acids Res. 43: D204-D212 |
| | creator | UniProt Consortium |
| | language | eng |
| | license | 3.0/ |
| | title | UniProt Protein C8V1L6 |
| | Version | UniProt release 2016_09 |
| | in dataset | Uni Prot Accessor/ |
| | contact point | contact |
| | description | Splicing factor u2af large subunit (AFU_orthologue AFUA_7G05310) |
| | distribution | Distribution D7566F52 C143 11E6 897C 26245D07C3DD |
| | | Distribution D75682F8 C143 11E6 897C 26245D07C3DD |
| | identifier | C8V1L6 |
| | keyword | Annotation |
| | | Aspergillus nidulans |
| | | Aspergillus |
| | | Functinal Annotation |
| | | GO |
| | | Gene Ontology |
| | | Proteins |
| | | RNA Processing |
| | landing page | uniprot.org |
| | language | en |
| | publisher | uniprot.org |
| | page | sparql |
| | | uniprot.org/ |
| | primary topic | C8V1L6 |
| C8V1L6 | ... | |
| Distribution D7566F52 C143 11E6 897C 26245D07C3DD | format | text/html |
| | type | Dataset |
| | | Distribution |
| | download URL | C8V1L6.html |
| Distribution D75682F8 C143 11E6 897C 26245D07C3DD | format | application/rdf+xml |
| | type | Dataset |
| | | dataset |
| | | Distribution |
| | download URL | C8V1L6.rdf |

**Figure 5  A representative (incomplete) portion of the output from resolving the MetaRecord Resource of the FAIR Accessor for record C8V1L6 (at http://linkeddata.systems/Accessors/UniProtAccessor/ C8V1L6), rendered into HTML by the Tabulator Firefox plugin.** The columns have the same meaning as in Fig. 4.

there are no FAIR Projectors for that dataset. This will be similar to the document returned by calling a FAIR MetaRecord URL in the Zenodo use case discussed in the earlier Methods section.

Calling HTTP GET on a MetaRecord Resource URL returns a document that include metadata elements and structure shown in Fig. 5. Note that Fig. 5 is not the complete MetaRecord; rather it has been edited to include only those elements relevant to the aspects of the interoperability infrastructure that have been discussed so far. More complete examples of the MetaRecord RDF, including the elements describing a Projector, are described in Figs. 6–9.

Many properties in this metadata document are similar to those at the higher level of the FAIR Accessor. Notably, however, the primary topic of this document is the UniProt record, indicating a shift in the focus of the document from the provider of the Accessor to the provider of the originating Data. Therefore, the values of these facets now reflect the authorship and contact information for that record. We do, recognize that MetaRecords are themselves scholarly works and should be properly cited. The MetaRecord includes the ''in dataset'' predicate, which refers back to the first level of the FAIR Accessor, thus this provides one avenue for capturing the provenance information for the MetaRecord. If additional provenance detail is required, we propose (but no not describe further here)

```
@prefix dc: <http://purl.org/dc/elements/1.1/>.
@prefix dcat: <http://www.w3.org/ns/dcat#>.
@prefix Uni: <http://linkeddata.systems/Accessors/UniProtAccessor/>.

Uni:C8V1L6
    dcat:distribution
        <#DistributionD7566F52-C143-11E6-897C-26245D07C3DD>,
        <#DistributionD75682F8-C143-11E6-897C-26245D07C3DD>;
<#DistributionD7566F52-C143-11E6-897C-26245D07C3DD>
    dc:format
        "text/html";
    a    dc:Dataset, dcat:Distribution;
    dcat:downloadURL
        <http://www.uniprot.org/uniprot/C8V1L6.html>.
<#DistributionD75682F8-C143-11E6-897C-26245D07C3DD>
    dc:format
        "application/rdf+xml";
    a    dc:Dataset, void:Dataset, dcat:Distribution;
    dcat:downloadURL
        <http://www.uniprot.org/uniprot/C8V1L6.rdf>.
```

**Figure 6** **Turtle representation of the subset of triples from the MetaRecord metadata pertaining to the two DCAT Distributions.** Each distribution specifies an available representation (media type), and a URL from which that representation can be downloaded.

| UniProt Protein C8V1L6 | | |
|---|---|---|
| | bibliographic Citation | The UniProt Consortium (2015). UniProt: a hub for protein information. Nucleic Acids Res. 43: D204-D212 |
| | creator | UniProt Consortium |
| | language | eng |
| | license | 3.0/ |
| | title | UniProt Protein C8V1L6 |
| | Version | UniProt release 2016_09 |
| | in dataset | Uni Prot Accessor/ |
| | contact point | contact |
| | description | Splicing factor u2af large subunit (AFU_orthologue AFUA_7G05310) |
| | distribution | Distribution9E275EC2 C1F6 11E6 8812 3E445D07C3DD |
| | | Distribution9E2771E6 C1F6 11E6 8812 3E445D07C3DD |
| | | Distribution9EFD1238 C1F6 11E6 8812 3E445D07C3DD |
| | | Distribution9EFD2458 C1F6 11E6 8812 3E445D07C3DD |
| | identifier | C8V1L6 |
| | keyword | Annotation |
| | | Aspergillus nidulans |
| | | Aspergillus |
| | | Functinal Annotation |
| | | GO |
| | | Gene Ontology |
| | | Proteins |
| | | RNA Processing |
| | landing page | uniprot.org |
| | language | en |
| | publisher | uniprot.org |
| | page | sparql |
| | | uniprot.org/ |
| | primary topic | C8V1L6 |

**Figure 7** **A portion of the output from resolving the MetaRecord Resource of the FAIR Accessor for record C8UZX9, rendered into HTML by the Tabulator Firefox plugin.** The columns have the same meaning as in Fig. 4. Comparing the structure of this document to that in Fig. 5 shows that there are now four values for the "distribution" predicate. An RDF and HTML representation, as in Fig. 5, and two additional distributions with URLs conforming to the TPF design pattern (highlighted).

```
@prefix dc: <http://purl.org/dc/elements/1.1/>.
@prefix dcat: <http://www.w3.org/ns/dcat#>.
@prefix rr: <http://www.w3.org/ns/r2rml#>.
@prefix ql: <http://semweb.mmlab.be/ns/ql#>.
@prefix rml: <http://semweb.mmlab.be/ns/rml#>.
@prefix Uni: <http://linkeddata.systems/Accessors/UniProtAccessor//>.
@prefix void: <http://rdfs.org/ns/void#>.
@prefix Uni: </cgi-bin/Accessors/UniProtAccessor/>.
@prefix FAI: <http://datafairport.org/ontology/FAIR-schema.owl#>.
@prefix core: <http://purl.uniprot.org/core/>.
@prefix edam: <http://edamontology.org/>.

Uni:C8V1L6
    dcat:distribution
        <#Distribution9EFD1238-C1F6-11E6-8812-3E445D07C3DD>,

<#Distribution9EFD1238-C1F6-11E6-8812-3E445D07C3DD>
    dc:format
        "application/rdf+xml", "application/x-turtle", "text/html";
    rml:hasMapping
        <#Mappings9EFD1238-C1F6-11E6-8812-3E445D07C3DD>;
    a    FAI:Projector, dc:Dataset, void:Dataset, dcat:Distribution;
    dcat:downloadURL
    <http://linkeddata.systems:3001/fragments?subject=http%3A%2F%2Fidentifiers%2Eorg
%2Funiprot%2FC8V1L6&predicate=http%3A%2F%2Fpurl%2Euniprot%2Eorg%2Fcore%2Fclassified
With>.

<#Mappings9EFD1238-C1F6-11E6-8812-3E445D07C3DD>
    rr:subjectMap
        <#SubjectMap9EFD1238-C1F6-11E6-8812-3E445D07C3DD>.
    rr:predicateObjectMap
        <#POMap9EFD1238-C1F6-11E6-8812-3E445D07C3DD>;

<#SubjectMap9EFD1238-C1F6-11E6-8812-3E445D07C3DD>
  rr:class edam:data_0896; rr:template "http://identifiers.org/uniprot/{ID}".

<#POMap9EFD1238-C1F6-11E6-8812-3E445D07C3DD>
    rr:objectMap
        <#ObjectMap9EFD1238-C1F6-11E6-8812-3E445D07C3DD>;
    rr:predicate
        core:classifiedWith.

<#ObjectMap9EFD1238-C1F6-11E6-8812-3E445D07C3DD>
  rr:parentTriplesMap <#SubjectMap29EFD1238-C1F6-11E6-8812-3E445D07C3DD>.

<#SubjectMap29EFD1238-C1F6-11E6-8812-3E445D07C3DD>
  rr:class ed:data_1176; rr:template "http://identifiers.org/taxon/{TAX}".
```

**Figure 8** **Turtle representation of the subset of triples from the MetaRecord metadata pertaining to one of the FAIR Projector DCAT Distributions of the MetaRecord shown in Fig. 7.** The text is colour-coded to assist in visual exploration of the RDF. The DCAT Distribution blocks of the two Projector distributions (black bold) have multiple media-type representations (red), and are connected to an RML Map (Dark blue) by the hasMapping predicate, which is a block of RML that semantically describes the subject, predicate, and object (green, orange, and purple respectively) of the Triple Descriptor for that Projector. This block of RML is schematically diagrammed in Fig. 2. The three media-types (red) indicate that the URL will respond to HTTP Content Negotiation, and may return any of those three formats.

| In UniProt | ```
http://purl.uniprot.org/uniprot/C8UZX9
        a
            http://purl.uniprot.org/core/Protein ;

    http://purl.uniprot.org/core/classifiedWith
        http://purl.obolibrary.org/obo/GO_0000462 .

http://purl.obolibrary.org/obo/GO_0000462
        a
            http://www.w3.org/2002/07/owl#Class
``` |
|---|---|
| After Projection | **http://identifiers.org/uniprot/**C8UZX9<br>       a<br>            **http://edamontology.org/data_0896 ;**<br><br>    http://purl.uniprot.org/core/classifiedWith<br>        http://purl.obolibrary.org/obo/GO_0000462 .<br><br>http://purl.obolibrary.org/obo/GO_0000462<br>       a<br>          **http://edamontology.org/data_1176** |

**Figure 9  Data before and after FAIR Projection.** Bolded segments show how the URI structure and the semantics of the data were modified, according to the mapping defined in the Triple Descriptor (data_0896 = "Protein report" and data_1176 = "GO Concept ID"). URI structure transformations may be useful for integrative queries against datasets that utilize the Identifiers.org URI scheme such as OpenLifeData (*González et al., 2014*). Semantic transformations allow integrative queries across datasets that utilize diverse and redundant ontologies for describing their data, and in this example, may also be used to add semantics where there were none before.

that this information could be contained in a separate named graph, in a manner akin to that used by NanoPublications (*Kuhn et al., 2016*).

The important distinctive property in this document is the "distribution" property, from the DCAT ontology. For clarity, an abbreviated document in Turtle format is shown in Fig. 6, containing only the "distribution" elements and their values.

There are two DCAT Distributions in this document. The first is described as being in format "application/rdf+xml", with its associated download URL. The second is described as being in format "text/html", again with the correct URL for that representation. Both are typed as Distributions from the DCAT ontology. These distributions are published by UniProt themselves, and the UniProt URLs are used. Additional metadata in the FAIR Accessor (not shown in Fig. 6) describes the keywords that relate to that record in both machine and human-readable formats, access policy, and license, allowing machines to more accurately determine the utility of this record prior to retrieving it.

Several things are important to note before moving to a discussion of FAIR Projectors. First, the two levels of the FAIR Accessor are not interdependent. The Container layer can describe relevant information about the scope and nature of a repository, but might not provide any further links to MetaRecords. Similarly, whether or not to provide a distribution within a MetaRecord is entirely at the discretion of the data owner. For sensitive data, an owner may chose to simply provide (even limited) metadata, but not provide any direct link to the data itself, and this is perfectly conformant with the FAIR guidelines. Further, when publishing a single data record, it is not obligatory to publish the

Container level of the FAIR Accessor; one could simply provide the MetaRecord document describing that data file, together with an optional link to that file as a Distribution. Finally, it is also possible to publish containers of containers, to any depth, if such is required to describe a multi-resource scenario (e.g., an institution hosting multiple distinct databases).

## The FAIR projector

FAIR Projectors can be used for many purposes, including (but not limited to) publishing transformed Linked Data from non-Linked Data; publishing transformed data from a Linked Data source into a distinct structure or ontological framework; load-management/query-management; or as a means to explicitly describe the ontological structure of an underlying data source in a searchable manner. In this demonstration, the FAIR Projector publishes dynamically transformed data, where the transformation involves altering the semantics of RDF provided by UniProt into a different ontological framework (EDAM).

This FAIR Projector's TPF interface is available at:

```
http://linkeddata.systems:3001/fragments
```

Data exposed as a TPF-compliant Resource require a subject and/or predicate and/or object value to be specified in the URL; a request for the all-variable pattern (blank, as above) will return nothing. How can a software agent know what URLs are valid, and what will be returned from such a request?

In this interoperability infrastructure, we propose that Projectors should be considered as DCAT Distributions, and thus TPF URLs, with appropriate parameters bound, are included in the distribution section of the MetaRecord metadata. An example is shown in Fig. 7, again rendered using Tabulator.

Note that there are now four distributions—two of them are the html and rdf distributions discussed above (Fig. 5). The two new distributions are those provided by a FAIR Projector. Again, looking at an abbreviated and simplified Turtle document for clarity (Fig. 8) we can see the metadata structure of one of these two new distributions.

Following the Triple Pattern Fragments behaviour, requesting the downloadURL with HTTP GET will trigger the Projector to restrict its output to only those data from UniProt where the subject is UniProt record C8V1L6, and the property of interest is "classifiedWith" from the UniProt Core ontology. The triples returned in response to this call, however, will not match the native semantics of UniProt, but rather will match the semantics and structure defined in the RML Mappings block. The schematic structure of this Mapping RML is diagrammed in Fig. 2. The Mappings describes a Triple where the subject will be of type `edam:data_0896` ("Protein record"), the predicate will be "`classifiedWith`" from the UniProt Core ontology, and the object will be of type `edam:data_1176` ("GO Concept ID").

Specifically, the triples returned are:

```
@prefix uni: <http://identifiers.org/uniprot/>.
@prefix obo: <http://purl.obolibrary.org/obo/>.
uni:C8V1L6 core:classifiedWith obo:GO_0000245, obo:GO_0045292 .
```

This is accompanied by a block of hypermedia controls (not shown) using the Hydra vocabulary (*Lanthaler & Gütl, 2013*; *Das, Sundara & Cyganiak, 2012*) that provide machine-readable instructions for how to navigate the remainder of that dataset—for example, how to get the entire row, or the entire column for the current data-point.

Though the subject and object are not explicitly typed in the output from this call to the Projector, further exploration of the Projector's output, via those TPF's hypermedia controls, would reveal that the Subject and Object are in fact typed according to the EDAM ontology, as declared in the RML Mapping. Thus, this FAIR Projector served data transformed from UniProt Core semantic types, to the equivalent data represented within the EDAM semantic framework, as shown in Fig. 9. Also note that the URI structure for the UniProt entity has been changed from the UniProt URI scheme to a URI following the Identifiers.org scheme.

The FAIR Projector, in this case, is a script that dynamically transforms data from a query of UniProt into the appropriately formatted triples; however, this is opaque to the client. The Projector's TPF interface, from the perspective of the client, would be identical if the Projector was serving pre-transformed data from a static document, or even generating novel data from an analytical service. Thus, FAIR Projectors harmonize the interface to retrieving RDF data in a desired semantic/structure, regardless of the underlying mechanism for generating that data.

This example was chosen for a number of reasons. First, to contrast with the static Zenodo example provided earlier, where this Accessor/Projector combination are querying the UniProt database dynamically. In addition, because we wished to demonstrate the utility of the Projector's ability to transform the semantic framework of existing FAIR data in a discoverable way. For example, in UniProt, Gene Ontology terms do not have a richer semantic classification than "owl:Class". With respect to interoperability, this is problematic, as the lack of rich semantic typing prevents them from being used for automated discovery of resources that could potentially consume them, or use them for integrative, cross-domain queries. This FAIR Accessor/Projector advertises that it is possible to obtain EDAM-classified data, from UniProt, simply by resolving the Projector URL.

## DISCUSSION

Interoperability is hard. It was immediately evident that, of the four FAIR principles, Interoperability was going to be the most challenging. Here we have designed a novel infrastructure with the primary objective of interoperability for both metadata and data, but with an eye to all four of the FAIR Principles. We wished to provide discoverable and interoperable access to a wide range of underlying data sources—even those in computationally opaque formats—as well as supporting a wide array of both academic and commercial end-user applications above these data sources. In addition, we imposed constraints on our selection of technologies; in particular, that the implementation should re-use existing technologies as much as possible, and should support multiple and unpredictable end-uses. Moreover, it was accepted from the outset that the trade-off between simplicity and power was one that could not be avoided, since a key objective was

to maximize uptake over the broadest range of data repositories, spanning all domains—this would be nearly impossible to achieve through, for example, attempting to impose a 'universal' API or novel query language. Thus, with the goal of maximizing global uptake and adoption of this interoperability infrastructure, and democratizing the cost of implementation over the entire stakeholder community—both users and providers—we opted for lightweight, weakly integrative, REST solutions, that nevertheless lend themselves to significant degrees of mechanization in both discovery and integration.

We now look more closely at how this interoperability infrastructure meets the expectations within the FAIR Principles.

**FAIR facet(s) addressed by the Container Resource:**

- **Findable**—The container has a distinct globally unique and resolvable identifier, allowing it to be discovered and explicitly, unambiguously cited. This is important because, in many cases, the dataset being described does not natively possess an identifier, as in our example above where the dataset represented the results of a query. In addition, the container's metadata describes the research object, allowing humans and machines to evaluate the potential utility of that object for their task.
- **Accessible**—the Container URL resolves to a metadata record using standard HTTP GET. In addition to describing the nature of the research object, the metadata record should include information regarding licensing, access restrictions, and/or the access protocol for the research object. Importantly, the container metadata exists independently of the research object it describes, where FAIR Accessibility requires metadata to be persistently available even if the data itself is not.
- **Interoperable**—The metadata is provided in RDF—a globally-applicable syntax for data and knowledge sharing. In addition, the metadata uses shared, widely-adopted public ontologies and vocabularies to facilitate interoperability at the metadata level.
- **Reusable**—the metadata includes citation information related to the authorship of the container and/or its contents, and license information related to the reuse of the data, by whom, and for what purpose.

**Other features of the Container Resource**

- **Privacy protection**—The container metadata provides access to a rich description of the content of a resource, without exposing any data within that resource. While a provider may choose to include MetaRecord URLs within this container, they are not required to do so if, for example, the data is highly sensitive, or no longer easily accessible; however, the contact information provided within the container allows potential users of that data to inquire as to the possibility of gaining access in some other way. As such, this container facilitates a high degree of FAIRness, while still providing a high degree of privacy protection.

**FAIR Facet(s) Addressed by the MetaRecord:**

- **Findable**—The MetaRecord URL is a globally-unique and resolvable identifier for a data entity, regardless of whether or not it natively possesses an identifier. The

metadata it resolves to allows both humans and machines to interrogate the nature of a data element before deciding to access it.

- **Accessible**—the metadata provided by accessing the MetaRecord URL describes the accessibility protocol and license information for that record, and describes all available formats.
- **Interoperable**—as with the Container metadata, the use of shared ontologies and RDF ensures that the metadata is interoperable.
- **Reusable**—the MetaRecord metadata should carry record-level citation information to ensure proper attribution if the data is used. We further propose, but do not demonstrate, that authorship of the MetaRecord itself could be carried in a second named-graph, in a manner similar to that proposed by the NanoPublication specification.

**Other features of the MetaRecord**

- **Privacy protection**—the MetaRecord provides for rich descriptive information about a specific member of a collection, where the granularity of that description is entirely under the control of the data owner. As such, the MetaRecord can provide a high degree of FAIRness at the level of an individual record, without necessarily exposing any identifiable information. In addition, the provider may choose to stop at this level of FAIRness, and not include further URLs giving access to the data itself.
- **Symmetry of traversal**—Since we predict that clients will, in the future, query over indexes of FAIR metadata searching for dataset or records of interest, it is not possible to predict the position at which a client or their agent will enter your FAIR Accessor. While the container metadata provides links to individual MetaRecords, the MetaRecord similarly provides a reference back "upwards" to its container. Thus a client can access repository-level metadata (e.g., curation policy, ownership, linking policy) for any given data element it discovers. This became particularly relevant as a result of the European Court of Justice decision (*Court of Justice of the European Union, 2016*) that puts the burden of proof on those who create hyperlinks to ensure the document they link to is not, itself, in violation of copyright.
- **High granularity of access control**—individual elements of a collection may have distinct access constraints or licenses. For example, individual patients within a study may have provided different consent. MetaRecords allow each element within a collection to possess, and publish, its own access policy, access protocol, license, and/or usage-constraints, thus providing fine-grained control of the access/use of individual elements within a repository.

**FAIR Facet(s) Addressed by the Triple Descriptors and FAIR Projectors:**

- **Findable**—Triple Descriptors, in isolation or when aggregated into FAIR Profiles, provide one or more semantic interpretations of data elements. By indexing these descriptors, it would become possible to search over datasets for those that contain data-types of interest. Moreover, FAIR Projectors, as a result of the TPF URI structure, create a unique URL for every data-point within a record. This has striking

consequences with respect to scholarly communication. For example, it becomes possible to unambiguously refer-to, and therefore "discuss" and/or annotate, individual spreadsheet cells from any data repository.

- **Accessible**—Using the TPF design patterns, all data retrieval is accomplished in exactly the same way—via HTTP GET. The response includes machine-readable instructions that guide further exploration of the data without the need to define an API. FAIR Projectors also give the data owner high granularity access control; rather than publishing their entire dataset, they can select to publish only certain components of that dataset, and/or can put different access controls on different data elements, for example, down to the level of an individual spreadsheet cell.
- **Interoperable**—FAIR Projectors provide a standardized way to export any type of underlying data in a machine-readable structure, using widely used, public shared vocabularies. Data linkages that were initially implicit in the datastore, identifiers for example, become explicit when converted into URIs, resulting in qualified linkages between formerly opaque data deposits. Similarly, data that resides within computationally opaque structures or formats can also be exposed, and published in a FAIR manner if there is an algorithm capable of extracting it and exposing it via the TPF interface.
- **Reusable**—All data points now possess unique identifiers, which allows them to be explicitly connected to their citation and license information (i.e., the MetaRecord). In this way, every data point, even when encountered in isolation, provides a path to trace-back to its reusability metadata.

**Other features of FAIR Projection**

- **Native formats are preserved**—As in many research domains, bioinformatics has created a large number of data/file formats. Many of these, especially those that hold "big data", are specially formatted flat-files that focus on size-efficient representation of data, at the expense of general machine-accessibility. The analytical tooling that exists in this domain is capable of consuming these various formats. While the FAIR Data community has never advocated for wholesale Interoperable representations of these kinds of data—which would be inefficient, wasteful, and lacking in utility—the FAIR Projector provides a middle-ground. Projection allows software to query the core content of a file in a repository prior to downloading it; for example, to determine if it contains data about an entity or identifier of interest. FAIR Projectors, therefore, enable efficient discovery of data of-interest, without requiring wasteful transformation of all data content into a FAIR format.
- **Semantic conversion of existing Triplestores**—It is customary to re-cast the semantic types of entities within triplestores using customized SPARQL BIND or CONSTRUCT clauses. FAIR Projectors provide a standardized, SPARQL-free, and discoverable way to accomplish the same task. This further harmonizes data, and simplifies interoperability.
- **Standardized interface to (some) Web APIs**—Many Web APIs in the biomedical domain have a single input parameter, generally representing an identifier for

some biochemical entity. FAIR Projectors can easily replace these myriad Web APIs with a common TPF interface, thus dramatically enhancing discoverability, machine-readability, and interoperability between these currently widely disparate services.

**Incentives and barriers to implementation**

Looking forward, there is every indication that FAIRness will soon be a requirement of funding agencies and/or journals. As such, infrastructures such as the one described in this exemplar will almost certainly become a natural part of scholarly data publishing in the future. Though the FAIR infrastructure proposed here may appear difficult to achieve, we argue that a large portion of these behaviours—for example, the first two layers of the Accessor—can be accomplished using simple fill-in-the-blank templates. Such templating tools are, in fact, already being created by several of the co-authors, and will be tested on the biomedical data publishing community in the near future to ensure they are clear and usable by this key target-audience.

Projection, however, is clearly a complex undertaking, and one that is unlikely to be accomplished by non-informaticians on their own. Transformation from unstructured or semi-structured formats into interoperable formats cannot be fully automated, and we do not claim to have fully solved the interoperability bottleneck. We do, however, claim to have created an infrastructure that improves on the *status quo* in two ways: first, we propose to replace the wasteful, one-off, "reuseless" data transformation activities currently undertaken on a daily basis throughout the biomedical community (and beyond), with a common, reusable, and machine-readable approach, by suggesting that all data transformations should be described in RML and transformed data exposed using TPF. Second, the solution we propose may, in many cases, partially automate the data transformation process itself. RML can be used, in combination with generic software such as RML Processor (http://github.com/RMLio) to actuate a data transformation over many common file formats such as CSV or XML. As such, by focusing on building RML models, *in lieu* of reuseless data transformation scripts, data publishers achieve both the desired data transformation, as well as a machine-readable interface that provides that transformed data to all other users. This may be incentivized even more by creating repositories of RML models that can be reused by those needing to do data transformations. Though the infrastructure for capturing these user-driven transformation events and formalizing them into FAIR Projectors does not yet exist, it does not appear on its surface to be a complex problem. Thus, we expect that such infrastructure should appear soon after FAIRness becomes a scholarly publishing requirement, and early prototypes of these infrastructures are being built by our co-authors.

Several communities of data providers are already planning to use this, or related FAIR implementations, to assist their communities to find, access, and reuse their valuable data holdings. For example, the Biobanking and Rare disease communities will be given end-user tools that utilize/generate such FAIR infrastructures to: guide discovery by researchers; help both biobankers and researchers to re-code their data to standard ontologies building on the SORTA system (*Pang et al., 2015*); assist to extend the MOLGENIS/BiobankConnect system

(*Pang et al., 2016*); add FAIR interfaces to the BBMRI (Biobanking and BioMolecular resources Research Infrastructure) and RD-connect national and European biobank data and sample catalogues. There are also a core group of FAIR infrastructure authors who are creating large-scale indexing and discovery systems that will facilitate the automated identification and retrieval of relevant information, from any repository, in response to end-user queries, portending a day when currently unused—"lost"—data deposits once again provide return-on-investment through their discovery and reuse.

## CONCLUSIONS

There is a growing movement of governing bodies and funding organizations towards a requirement for open data publishing, following the FAIR Principles. It is, therefore, useful to have an exemplar "reference implementation" that demonstrates the kinds of behaviours that are expected from FAIR resources.

Of the four FAIR Principles, Interoperability is arguably the most difficult FAIR facet to achieve, and has been the topic of decades of informatics research. Several new standards and frameworks have appeared in recent months that addressed various aspects of the Interoperability problem. Here, we apply these in a novel combination, and show that the result is capable of providing interoperability between formerly incompatible data formats published anywhere on the Web. In addition, we note that the other three aspects of FAIR—Findability, Accessibility, and Reusability—are easily addressed by the resulting infrastructure. The outcome, therefore, provides machine-discoverable access to richly described data resources in any format, in any repository, with the possibility of interoperability of the contained data down to the level of an individual "cell". No new standards or APIs were required; rather, we rely on REST behaviour, with all entities being resources with a resolvable identifier that allow hypermedia-driven "drill-down" from the level of a repository descriptor, all the way to an individual data point in the record.

Such an interoperability layer may be created and published by anyone, for any data source, without necessitating an interaction with the data owner. Moreover, the majority of the interoperability layer we describe may be achieved through dynamically generated files from software, or even (for the Accessor portion) through static, manually-edited files deposited in any public repository. As such, knowledge of how to build or deploy Web infrastructure is not required to achieve a large portion of these FAIR behaviours.

The trade-off between power and simplicity was considered acceptable, as a means to hopefully encourage wide adoption. The modularity of the solution was also important because, in a manner akin to crowdsourcing, we anticipate that the implementation will spread through the community on a needs-driven basis, with the most critical resource components being targeted early—the result of individual researchers requiring interoperable access to datasets/subsets of interest to them. The interoperability design patterns presented here provide a structured way for these individuals to contribute and share their individual effort—effort they would have invested anyway—in a collaborative manner, piece-by-piece building a much larger interoperable and FAIR data infrastructure to benefit the global community.

# ACKNOWLEDGEMENTS

In January 2014 the Lorentz Center hosted the 'Jointly Designing a Data FAIRport' workshop. This workshop was organized by Barend Mons in collaboration with and co-sponsored by the Lorentz center, The Dutch Techcentre for Life Sciences/ELIXIR-NL and the Netherlands eScience Center. The workshop led to the formalization of FAIR principles and subsequently to the formation of a FAIR Skunkworks team and a FAIR Data engineering team. We thank Barend Mons for critical discussions leading up to this article. We would also like to thank the UniProt RDF and SPARQL team at the Swiss-Prot group of the SIB Swiss Institute of Bioinformatics for their advice and assistance. We would like to acknowledge the advice and feedback from the leaders and participants of BioHackathon 2016, hosted by the Integrated Database Project (Ministry of Education, Culture, Sports Science and Technology, Japan), the National Bioscience Database Center (NBDC—Japan), and the Database Center for Life Sciences (DBCLS—Japan).

### Funding

The lead author is supported by the Fundacion BBVA + UPM Isaac Peral programme, and the Spanish Ministerio de Economía y Competitividad grant number TIN2014-55993-R. Additional support for FAIR Skunkworks members comes from European Union funded projects ELIXIR-EXCELERATE (H2020 no. 676559), ADOPT BBMRI-ERIC (H2020 no. 676550) and CORBEL (H2020 no. 654248). Portions of this work have been funded by Netherlands Organisation for Scientific Research (Odex4all project), Stichting Topconsortium voor Kennis en Innovatie High Tech Systemen en Materialen (FAIRdICT project), BBMRI-NL, RD-Connect and ELIXIR (Rare disease implementation study FP7 no. 305444). UniProt is mainly supported by the National Institutes of Health (NIH), National Human Genome Research Institute (NHGRI) and National Institute of General Medical Sciences (NIGMS) grant U41HG007822. Swiss-Prot activities at the SIB are supported by the Swiss Federal Government through the State Secretariat for Education, Research and Innovation SERI. There was no additional external funding received for this study. The funders had no role in study design, data collection and analysis, decision to publish, or preparation of the manuscript.

### Grant Disclosures

The following grant information was disclosed by the authors:
Fundacion BBVA + UPM Isaac Peral programme.
Spanish Ministerio de Economía y Competitividad: TIN2014-55993-R.
European Union funded projects ELIXIR-EXCELERATE: H2020 no. 676559.
ADOPT BBMRI-ERIC: H2020 no. 676550.
CORBEL: H2020 no. 654248.
Netherlands Organisation for Scientific Research.
FAIRdICT project.

National Institutes of Health (NIH).
National Human Genome Research Institute (NHGRI).
National Institute of General Medical Sciences (NIGMS): U41HG007822.
Swiss Federal Government.

## Competing Interests

The authors declare there are no competing interests.

## Author Contributions

- Mark D. Wilkinson conceived and designed the experiments, performed the experiments, analyzed the data, wrote the paper, prepared figures and/or tables, performed the computation work, reviewed drafts of the paper.
- Ruben Verborgh conceived and designed the experiments, performed the experiments, analyzed the data, contributed reagents/materials/analysis tools, wrote the paper, performed the computation work, reviewed drafts of the paper.
- Luiz Olavo Bonino da Silva Santos, Fleur D.L. Kelpin, Arnold Kuzniar and Anand Gavai conceived and designed the experiments, analyzed the data, reviewed drafts of the paper.
- Tim Clark, Alasdair J.G. Gray, Erik M. van Mulligen and Paolo Ciccarese conceived and designed the experiments, reviewed drafts of the paper.
- Morris A. Swertz, Erik A. Schultes, Mark Thompson and Rajaram Kaliyaperumal conceived and designed the experiments, analyzed the data, wrote the paper, reviewed drafts of the paper.
- Jerven T. Bolleman analyzed the data, contributed reagents/materials/analysis tools, reviewed drafts of the paper, fixed the demonstrative query, clarified the semantics of UniProt, corrected erroneous ontological annotations;.
- Michel Dumontier conceived and designed the experiments, performed the experiments, analyzed the data, wrote the paper, reviewed drafts of the paper.

## Data Availability

The manuscript describes a set of practices and behaviors that combine third-party technologies and standards in a novel manner. This does not (necessarily) require novel, dedicated software, and therefore a repository is not provided. The paper uses only public data for its demonstration, and the query to retrieve that data from-source is provided in the manuscript text (the curator of that data is UniProt, the data is being used/republished with their explicit permission, and a member of their team is a co-author on the manuscript).

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
