# Peer review of "Interoperability and FAIRness through a novel combination of Web technologies"

_PeerJ Computer Science, doi:10.7717/peerj-cs.110_

## Round 0.1 · original submission · Major Revisions

Please take into account all the comments provided by the three reviewers, and include a letter explaining how have you addressed all these suggestions.

·

Basic reporting

The article covers all the points required. The article is written in english and I guess that has been submitted using the available templates (not sure about that, I currently can't access the templates). The figures have enough resolution and are easy to read and are clear. I consider the submission self-contained and the raw data is within the paper or available online in the URLs provided.

Experimental design

The submission is original. As far as I know this paper has not been submitted elsewhere. The research question/aim is clearly defined and it is relevant and meaningful. Since it is provided a set of methodological steps/patterns they are all discussed with enough rigor and provided with useful examples. The methods are described with sufficient information (in most of the sections; in those that I consider that more explanations are needed, I've pointed out.

Validity of the findings

In this case we are talking about a set of patterns for data discovery, hence, classical validation from statistical point of view or similar according to the scientific classical guideliness are not applicable. However, the work presented is valid and the authors have demonstrated its value using a good explanation and also providing valid examples.

Additional comments

The paper provides a set of patterns for data discovery, accessibility, transformation and integration to be implemented in data repositories (general or specific ones; authors focused their work on life sciences) by means of following FAIR data principles. The overall goal of the paper is easy to understand and it is very interesting. The paper is easy to read and to follow the main ideas.
However, I've some specific comments and suggestions that I think that could be useful for the authors. I also potentially disagree with some of the opinions. Please note that my comments I wrote my comments as I was reading so maybe some comments in fact were solved/clarified later.
Authors make use of relatively new concepts within the semantic web area: RML and TPF. I think that more information in form of examples could be useful for the non-expert users to improve the understanding of the paper as well as these particular concepts.
In line 306 there is a sentence that said: “the lack of a novel API means that the information is accessible to generic Web-crawling agents, and may also be processed if that agent “understands” the vocabularies used”. Well, that it is clear and that is some of the main problems to deal with in the context of automatic retrieval, understanding and interoperability within the semantic web. My question is, is possible to address this problem with not-known vocabularies or remains as an open problem? What are the vocabularies to be used or where can they be retrieved? Just please explain a bit more regarding this.
In line 328 authors claim about the support of a multitude of source data formats (including for example, Excel, CSV, JSON, …) but all of this formats are structured formats (even Excel with its own API). However, is it possible to deal with binary files such as PDF that also need specific APIs to get the data and the result will be given, normally, in unstructured text? In line 336 the authors claim that the approach selected was based on the premise that data, in any format, could be metamodeled as a first step towards interoperability, and honestly I’ve serious doubts that this could be really done always, mainly based on the type of object (data source – file) to be used and/or the internal structure. More information regarding this possible limitation would be helpful to understand your approach.
The idea of the FAIR projectors is interesting itself, but I couldn’t catch the idea that you are proposing. In line 446 authors said: “what is required is a universally-applicable way of retrieving data from any transformation script (or any data source), without inventing a new API. We now describe our suggestion for how to achieve this behaviour, and we refer to such transformation tools as FAIR projectors”. What exactly does the FAIR projectors tools? How are they able to universally transform the data retrieved from any transformation script or data source to the appropriate triples? I’m sorry but I’m a bit lost over here.
The examples provided regarding the FAIR Accessor from the Container Resource URL are quite nice. My only doubt is regarding when you call to a MetaRecord Resource. I tried to do the GET petition to the Container Resource and I successfully obtained the FAIR Accessor. From the FAIR Accessor I got what I understand/think is a MetaRecord (the first record in fact provided as result from the query provided in the Accessor (http://linkeddata.systems/Accessors/UniProtAccessor/C8UZX9 ). According to Figure 5, the MetaRecord should provide me some information for record C8UZX9, and it is true, this information is provided. But this information is provided after lot of triples that honestly I don’t know what they mean. Some of the triples seems to be related with the type of information obtained (taxon, organism, …) but some others are like this one:
<rdf:Description xmlns:ns1="http://semweb.mmlab.be/ns/rml#" xmlns:ns2="http://www.w3.org/ns/r2rml#" rdf:about="http://datafairport.org/local/MappingsFCCC8188-99F0-11E6-A61C-84165D07C3DD">
<ns1:logicalSource rdf:resource="http://datafairport.org/local/SourceFCCC8188-99F0-11E6-A61C-84165D07C3DD"/>
<ns2:predicateObjectMap rdf:resource="http://datafairport.org/local/POMapFCCC8188-99F0-11E6-A61C-84165D07C3DD"/>
<ns2:subjectMap rdf:resource="http://datafairport.org/local/SubjectMapFCCC8188-99F0-11E6-A61C-84165D07C3DD"/>
</rdf:Description>

<rdf:Description xmlns:ns1="http://www.w3.org/ns/r2rml#" rdf:about="http://datafairport.org/local/ObjectMapFCCC90B0-99F0-11E6-A61C-84165D07C3DD">
<ns1:parentTriplesMap rdf:resource="http://datafairport.org/local/SubjectMap2FCCC90B0-99F0-11E6-A61C-84165D07C3DD"/>
</rdf:Description>
And the URLs contained over these triples are not, currently, resolvable being difficult to get more information about them.
Ok, now I see that this information is part of the projector. It would be interesting to make some reference to this when the MetaRecord examples are provided because if the reader tries to get the data at the same time that he is reading, probably could get lost. In any case, I suppose that the maps should resolve, and they are not currently doing it.
Regarding the transformation performed by the projector, I’m not sure if I understand completely the aim. As far as I saw in the example, the projector is trying to transform the native RDF semantics from UniProt to EDAM. However, the RML is describing a triple where the subject type is ‘Protein Record’ and the object ‘GO Concept ID’. Where is exactly this transformation of semantics? As far as I understand is returning an upper-class type for both elements of the triple (subject and object) that match with the predicate considered (classifiedWith). Is that the aim? I was wondering to replace with the equivalent elements (if applicable) for those instances in EDAM. What is the reason of performing this transformation? Please clarify. I wonder if in this case it has been just done with this upper class because it is not possible to do it in other way, but in other cases can be achieved.
As a final comment, I would like to introduce some kind of more descriptive workflow figure with all the elements involved. At the end it is difficult to understand the relation of the different elements provided with the previous one, and probably this kind of figure would help to have a better understanding.
Regarding the conclusions, although I consider the initiative very interesting, I think that would be very difficult to achieve the goals covet by the authors. It is necessary the development of more easy and friendly tools that allows the researchers to not waste too much time in thinking how to deploy their data according to FAIR principles (even if journals are requiring it..), but that’s just my opinion.
Typos:
- Please check Dimou et al. reference (it always appears without the year in the text as well as in the references list).
- In “Metadata Interoperability – The “FAIR Accessor” and the Linked Data Platform” section please ensure that you define the acronym of Linked Data Platform (you describe the term and then use the acronym, but you don’t relate them). Also, in this first paragraph of the section there is a statement regarding LDP definition from https://www.w3.org/TR/ldp/ that has been written in italic and between quotes but there is no reference. I think that reference should be included.
- Line 843: double efficient

Reviewer 2 ·

Basic reporting

"No Comments"

Experimental design

The paper lacks serious discussion about previous work related to the topic. How the authors place their solution compared to previous projects. Was it possible that the FAIR principles would be achieved by a minor tweaking of current systems?

Validity of the findings

"No Comments".

Additional comments

Review of the paper titled “Interoperability and FAIRness through a novel combination of web technologies”

The paper address an important topic regarding the access and retrieval of data from heterogonous resources. The paper uses on the shelf technologies to achieve this task.
I have the following concerns regarding the paper.
- The paper need to be re-organized. I had to jump from one section to the other to understand the topic. The authors concern to address the FAIR principles lead to interruption of flow of ideas. Starting with the methodology section without concrete example made it difficult to follow the contribution. A concrete example given in the introduction would have been enough to show the challenge and how the solution would work.
- I could not find serious discussion about previous work related to the topic. How the authors place their solution compared to previous projects. Was it possible that the FAIR principles would be achieved by a minor tweaking of current systems?
- The suggested method is not really easy and straightforward to implement. What is the technological requirements at both the server and client side to achieve this? How this can be simplified?
- What makes this method appealing for non-computer scientists? The use of different confusing technologies contradicts the concept of minimal design. Is there any solution to simplify this?

Reviewer 3 ·

Basic reporting

The paper is well-written and the structure is clear and appropriate. The authors clearly highlight the relevance of the topic in both the abstract and the introduction, providing the reader with a precise context regarding the application domain.

Focusing on the Introduction, the authors might include more information about existing alternatives and previous attempts to provide interoperability and easy integration of data repositories for bioinformatics or other related disciplines. Actually, it is briefly mentioned in the Discussion (lines 709-717), but I would suggest providing further information in this regard as part of the motivation at the beginning of the paper.

Similarly, the last paragraph of the Introduction (l. 156-162) could be extended in order to focus the scope of the paper. For instance, a brief introduction to the basic foundations of the selected technologies and how them are applied to address the challenge of data integration and availability might be useful in order to provide the reader with a more detailed overview of the proposed solution. I think that both aspects (background and contribution) can strengthen an already good introduction.

In general, the figures provide illustrative examples of the concepts surrounding the different technologies. Only two of them need some changes. On the one hand, Figure 1 is located at the beginning of Section 2 (methods) but it is not explained in the context of that section. It is only referred in Section 3 (results), even though the figure does not contain specific information about the case study. Moreover, DCAT has not been explained at this point, so it can be a bit confusing to the reader. On the other hand, Figure 2 assumes that the reader have strong knowledge on the "Triple Descriptor" representation. In fact, it is a major issue throughout the entire paper, as I mention later (please, see general comments). I would suggest including a better explanation of the role of "subject", "predicate" and "object" (in general) and the meaning of the specific "maps" that appear in Figure 2.

Experimental design

The paper does not provide experimental results as such, but instead the authors present an illustrative example. The case study serves to complement the technological foundations of the proposed system, now explaining the meaning of specific data and metadata fields.

The introduction to the results might be improved if a brief description or external references to UniProt and EDAM are provided. In this sense, the paper assumes that the reader has high expertise on several semantic aspects (protein records, ontologies...), which might hamper its readability. In the same lines, the code example shown in lines 508-524 would be rather difficult to understand by readers without knowledge on query languages. It is a particular issue that clearly needs revision, as the role of SPARQL was not explained in Section 2 (methods).

Were these changes applied, the paper can reach the necessary compromise between domain experts and those readers closer to the computing field, being more self-contained as well. More details about the scope of the case study and the used materials could be provided for the sake of reproducibility.

Validity of the findings

The validity of the findings may be compromised due to the lack of experimental results. In this sense, providing public access to the case study should be considered, if possible. Even though the implementation is not complete, it would be really interesting to let readers "play" with the data and navigate across the different FAIR profiles. It will be also in line with the policy of the journal that promotes sharing data and materials.

As part of the conclusions, the authors might discuss the challenges behind the final implementation of the proposed architecture. It includes dealing with huge volume of data and guaranteeing quality of service, among others. Similarly, current limitations and lines of future work should be specified.

Additional comments

A major concern when reading the paper is that the authors do not explicitly mention the current state of the implementation. Although it looks promising and highly valuable to the research community, it seems that the solution is not fully functional yet. As I see it, the manuscript is a position paper presenting a first approximation to the operational solution. I believe that this is not a problem for acceptance, but the paper should not be ambiguous in this point.

It is worth mentioning that the paper provides a comprehensive justification of the technologies selected to build the system. It really helps to present the general idea behind the system and then go into more specific details. The only weak point of this approach I have found is that concepts like "subject", "predicate" and "object" in RDF/RML were not described somewhere in between. For instance, lines 361-365 mention some restrictions about "objects" and "subjects" in the context of RML maps, but the general meaning of these terms seems to come from RDF, which might be unknown for the reader. Similarly, the acronym [S,P,O] is used later (l. 452) to refer to triple pattern fragments (TPF), but understanding its relation to the aforementioned terms is not straightforward.

I find really interesting the idea of multiple FAIR profiles to describe the same data, so that the system can also integrate different data formats. As I understand it, different data users/providers could create new profiles for the same data resource, so a question here is whether the system can deal with possible inconsistencies between the information provided by different profiles. Similarly, some kind of control regarding who publishes new content and what is being published might be required in the future.

Following with this topic, the collaborative approach opens new possibilities to enrich the system, such as the inclusion of opinions about the quality of the data or a compilation of studies based on them.

Some minor comments:

- There is no publication year for reference (Dimou et al.) (l. 231, and others)

- Please, provide the full name or an external reference for acronyms: FoaF (l.551), SKOS (l.556) and BBMRI (l.883).

- Typos: furrther (l.582), "efficient" appears twice (l. 843).

- Line 634, the reference to Figure 6 seems to be incorrect, should it be Figure 7?

- Mixture of British and American spelling: behaviors (l. 58 and others) vs. behaviours (l. 895, l.908)

---

## Round 0.2 · accepted · Accept

Congratulations for the good job. I hope you continue working with us in future times.

·

Basic reporting

The article covers all the points required. The article is written in english and seems to have been submitted using the available templates .The figures have enough resolution and are easy to read and are clear. I consider the submission self-contained and the raw data is within the paper or available online in the URLs provided.

Experimental design

The submission is original. As far as I know this paper has not been submitted elsewhere. The research question/aim is clearly defined and it is relevant and meaningful. Since it is provided a set of methodological steps/patterns they are all discussed with enough rigor and provided with useful examples.

The methods have been suffciently described in comparison with the previous versión. A replication of this work seems to be not applicable based on the paper aim, but possible with the original data.

Validity of the findings

In this case we are talking about a set of patterns for data discovery, hence, classical validation from statistical point of view or similar according to the scientific classical guideliness are not applicable. However, the work presented is valid and the authors have demonstrated its value using a good explanation and also providing valid examples.

Additional comments

The authors have taken into account my comments and have provided a good explanation to my doubts and inquiries. From my point of view and after also taking a look to the other reviewers from my side the paper it is currently ok and can be published. I really appreciate the efforts made by the authors to signifcantly improve the manuscript.

Reviewer 2 ·

Basic reporting

This version of the paper is an improved version of previous submission. The authors could address the comments in a good way.

Experimental design

NA

Validity of the findings

NA

Additional comments

This version of the paper addresses the comments given in previous submission. There are still challenges intrinsic in the topic addressed but this beyond the solution the authors provide..

Reviewer 3 ·

Basic reporting

No additional comments.

Experimental design

No additional comments.

Validity of the findings

No additional comments.

Additional comments

The authors have addressed all the comments and suggestions. From my point of view, the paper can be accepted for publication in its current state. To summarise the observed changes:

Introduction: the section has been improved according to my comments. It now provides sufficient information about the context and the contribution.

Terminology and related concepts: the text is now better organised and new references have been included, so every technical term is briefly introduced before its use and/or properly cited. I understand (and agree) that adding long definitions would hamper readability. After the minor changes, the manuscript does a proper balance between the biological and the computing perspectives.

URLs: Thanks for the explanation about the current state of the infrastructure. Probably it was a problem from my side, because I can access now to the UniProtAccessor. However, I cannot reach all the mentioned resources, e.g. https://zenodo.org/deposit/47641/, http://linkeddata.systems:3001/fragments
and linkeddata.systems in Figure 8. I suggest checking all the (new) URLs before submitting the final manuscript.

A minor comment: HTML and RDF are shown in lowercase in line 845.